# Estimación de la edad de la muerte basada en aprendizaje conjunto y regresión simbólica a partir de múltiples anotaciones

## Abstract

Este estudio aborda el problema de la estimación semiautomática de la edad de la muerte a partir de la sínfisis púbica, una tarea crucial aunque compleja en antropología forense. Su precisión depende directamente de la calidad del etiquetado de distintos rasgos en el hueso púbico desarrollado por los forenses, el cual está afectado por una incertidumbre inherente a su definición.

Dado que la interpretabilidad es un requisito obligatorio, proponemos un enfoque en el que el diseño del modelo se basa en aprendizaje evolutivo, considerando la programación genética para resolver el problema mediante regresión simbólica. Además, se usa el aprendizaje conjunto para abordar los retos que plantean el ruido, la incertidumbre y las anotaciones contradictorias inherentes a los datos recogidos de múltiples sujetos. El aprendizaje conjunto proporciona un enfoque eficaz para superar estos retos, ya que facilita la obtención de consenso mediante la toma de decisiones y la fusión de información. De ahí que se formen comités de observadores, compuestos por múltiples especialistas forenses con diferentes niveles de experiencia y conocimiento que proporcionan anotaciones alternativas.

Se prueban varias configuraciones de modelos de conjunto que combinan diferentes estimadores base y operadores de agregación para evaluar la precisión y la fiabilidad de sus estimaciones de la edad de la muerte. Se comparan con modelos entrenados con anotaciones individuales, mostrando una mejora en su rendimiento. Los resultados obtenidos también ponen de relieve las ventajas de incorporar diversas perspectivas para abordar las complejidades asociadas a la variabilidad humana y las evaluaciones anatómicas.

## 1. Introduction

Estimar la edad tanto de los vivos como de los muertos es una tarea crucial en Antropología Forense (AF) [1]. Específicamente, la estimación de la edad de la muerte implica el examen de rasgos específicos como la apariencia, la morfología y los patrones de osificación presentes en los restos óseos de los sujetos. La sínfisis púbica es un hueso ampliamente utilizado para esta tarea debido a su alta fiabilidad. El método pionero fue propuesto en 1920 por Todd [2], y aunque hasta la fecha se han propuesto muchos otros enfoques, sigue siendo una técnica relevante e influyente. La metodología estándar implica el análisis visual de la morfología de la superficie ósea y la categorización (basada en expertos humanos) de diferentes características o rasgos para determinar un rango de edad aproximado [2] o un valor numérico concreto [3].

Por lo tanto, la estimación de la edad de la muerte plantea un reto complejo caracterizado por la incertidumbre inherente y la susceptibilidad al sesgo humano, a menudo exacerbado por los pocos

Submitted to XVI Congreso Español de Metaheurísticas, Algoritmos Evolutivos y Bioinsiprados (MAEB 2025). Do not distribute.

datos disponibles. Esto ha aumentado recientemente el interés por el desarrollo de métodos semi [4,5] o totalmente automáticos [6,7] para ayudar al antropólogo forense. Es obligatorio obtener modelos interpretables y explicables, ya que este tipo de métodos suelen aplicarse en entornos médico-legales y los forenses deben confiar en ellos. Las aproximaciones más interpretables existentes en la literatura siguen basándose en la observación manual del hueso para caracterizar los diferentes rasgos de la sínfisis púbica implicados en la toma de decisiones. Sin embargo, esto se considera una tarea difícil debido a la incertidumbre que rodea a las definiciones de los rasgos esqueléticos [8]. En consecuencia, la metodología para estimar la edad de la muerte resulta ser muy sensible a la calidad de las anotaciones humanas de las variables, que dependen sobre todo de la experiencia y los conocimientos del especialista forense.

La regresión simbólica (RS) surge como un enfoque ideal para abordar las complejidades de las anotaciones incoherentes. La capacidad de la RS para extraer expresiones matemáticas de los datos le permite adaptarse y aprender de diversos conjuntos de anotaciones, capturando los patrones subyacentes y modelando relaciones únicas ocultas en los datos. En este trabajo, se encarga a una serie de especialistas forenses que examinen visualmente y extraigan una serie de rasgos óseos siguiendo una metodología basada en el método de Gilbert y McKern [3], que mejora la propuesta canónica de Todd estimando directamente un valor numérico para la edad de la muerte a partir de una combinación de evaluaciones parciales realizadas sobre cada rasgo óseo del pubis. A continuación, las anotaciones se utilizan para aprender un estimador de RS basado en la programación genética (PG). La interpretabilidad inherente de la PG, derivada de su naturaleza simbólica, permite el diseño de modelos de estimación de la edad de la muerte flexibles, precisos y semiautomáticos, explorando un espacio de modelos más extenso [9].

En el contexto del aprendizaje automático (AA), el campo conocido como aprendizaje a partir de multitudes (*learning from crowds*) suele implicar el aprovechamiento de entradas o anotaciones de múltiples sujetos para realizar predicciones/clasificaciones o tomar decisiones. A menudo, estas anotaciones son ruidosas o contradictorias, ya que sufren de sesgos humanos y dependen de la dificultad de la muestra y el nivel de habilidad o competencia de cada observador [10]. Por lo tanto, este campo de investigación tiene una relación directa con la incertidumbre inherente presente en el proceso de etiquetado de los rasgos de la sínfisis púbica en nuestro dominio. Nuestra hipótesis es que una aplicación adecuada del aprendizaje a partir de multitudes nos permitirá superar el problema del etiquetado de rasgos de la sínfisis púbica en la estimación de la edad de la muerte, logrando modelos más robustos a pesar de la posible existencia de anotaciones de baja calidad.

Aunque existen diferentes aproximaciones para tratar escenarios con múltiples anotadores, en este trabajo seguimos un enfoque de fusión de información basado en aprendizaje conjunto. Utilizando modelos de AA diseñados con PG [11], este estudio busca mejorar la robustez del método introduciendo comités de observadores [12], con el objetivo de generar modelos predictivos más efectivos. El uso de algoritmos evolutivos (AEs) y aprendizaje conjunto es un tema activo de investigación [13], impulsado por los beneficios reconocidos de los modelos de conjunto en términos de eficiencia computacional, rendimiento predictivo y especialización del espacio de características. En particular, este trabajo se centrará en el proceso de toma de decisiones para alcanzar un consenso. Para ello, evaluaremos diferentes operadores de agregación para combinar los modelos base [14].

El resto de este trabajo se estructura de la siguiente forma. La sección 2 explora el problema de estimación de la edad de la muerte y los fundamentos de la RS con PG. La sección 3 detalla las particularidades de la muestra de datos y el procedimiento de anotación. También introduce la metodología adoptada para abordar nuestro problema, describiendo la configuración algorítmica del método de PG empleado y los diferentes diseños de modelos de conjunto para fusionar la información. Por último, en la sección 4 se presentan los experimentos desarrollados y su análisis, y en la sección 5 se exponen algunas observaciones finales.

## 2. Preliminares

### 2.1. Estimación de la edad de la muerte a partir de la sínfisis púbica

La AF juega un papel decisivo en las investigaciones criminales y los procesos judiciales, especializándose en el análisis de restos óseos para la identificación humana. Mediante un proceso preliminar conocido como perfilado biológico, esta disciplina proporciona información crítica sobre características biológicas clave como la edad, el sexo, la ascendencia, la estatura y evidencias relacio-

nadas con la causa de la muerte. En particular, la edad de la muerte es un factor crucial que reduce significativamente el alcance de las posibles coincidencias durante el proceso de identificación.

Los métodos empleados por los forenses para abordar el problema de la estimación de la edad de la muerte son obsoletos [2, 3, 15]. En los últimos años, se ha producido un notable aumento del interés en el campo de la AF por el desarrollo de métodos precisos, robustos y automáticos para la estimación de la edad de la muerte. En particular, los avances recientes se han centrado en la automatización completa del proceso, integrando la Visión por Ordenador y el AA. Esta forma de proceder reduce la dependencia de los antropólogos forenses, a costa de dar lugar a métodos excesivamente intrincados y opacos. En consecuencia, la mayoría de los enfoques contemporáneos se basan en la extracción automática de características a partir de modelos óseos escaneados en 3D [6, 7]. Teniendo en cuenta la limitada interpretabilidad de las metodologías recientes y su compleja aplicabilidad, existe una búsqueda continua de enfoques semiautomáticos destinados a mitigar los sesgos.

En 2022, Gámez et al. [5] desarrollaron un enfoque de aprendizaje transparente basado en reglas evolutivas para la estimación de la edad de la muerte a partir de la sínfisis púbica, incorporando el conocimiento experto del método de Todd e imitando su modo de funcionamiento. El objetivo del AA era abordar un problema de clasificación ordinal, que implicaba la asignación de una de las diez fases de Todd que delimitan un rango de edad estimado para cada hueso púbico.

### 2.2. El método de Gilbert-McKern para estimar la edad de la muerte con la sínfisis púbica

En este estudio presentamos un nuevo método interpretable y semiautomático inspirado en el método de Gilbert y McKern [3]. La propuesta de Gilbert y McKern pretendía mejorar el método de Todd con dos modificaciones clave. A los rasgos del hueso púbico se les asignaba un valor numérico dentro del rango $\{0, 5\}$, valores que luego se agregaban para estimar directamente la edad de la muerte numérica. La propuesta consideraba tres zonas diferentes del hueso (componentes): la demi-facial dorsal, la rampante ventral y el borde sinfisal de la sínfisis púbica.

La agregación de los valores asignados a cada componente daba como resultado un número directamente asociado a un valor de edad de la muerte. En un estudio posterior [16], McKern llegó a la conclusión de que sus resultados superaban a los del método pionero de Todd.

### 2.3. Regresión Simbólica con Programación Genética

La RS es una forma de análisis de regresión que pretende representar la relación subyacente de los datos sin conocimiento previo de la expresión matemática resultante. Una ventaja clave de la RS reside en su imparcialidad, ya que no se ve afectada por los sesgos del diseñador del modelo ni por las incertidumbres en el conocimiento del dominio. Muchos enfoques utilizan comúnmente EAs [17] como potentes técnicas de optimización global para navegar por el intrincado espacio de búsqueda de las expresiones matemáticas. La PG [18] se erige como un enfoque clásico que se alinea con los principios básicos de los algoritmos genéticos, basándose en una gramática basada en árboles para representar a los individuos. Así, los operadores genéticos se aplican tanto a los nodos del árbol como a los subárboles, con las expresiones matemáticas codificadas como nodos del árbol y los operandos representados como nodos terminales.

La RS y la PG son especialmente adecuadas para problemas en los que la interpretabilidad y la transparencia son cruciales [9], como el actual. Estos enfoques destacan por proporcionar no sólo modelos predictivos, sino también expresiones comprensibles para el ser humano que ofrecen información sobre las relaciones subyacentes en los datos. Además, los modelos de PG resultantes suelen ser más intuitivos y fáciles de aplicar que los complejos modelos de caja negra, lo que facilita su aplicación práctica en los procedimientos forenses.

## 3. Materiales y Métodos

### 3.1. Descripción del conjunto de datos

El conjunto de datos empleado ha sido la colección de sínfisis púbicas esqueléticas conservadas en el laboratorio de Antropología Física de la Universidad de Granada (España). Los estudios de autopsia se vienen recopilando desde 1991 conjuntamente con el Instituto de Medicina Legal y Ciencias Forenses de Granada, dando lugar a uno de los mayores conjuntos de datos del mundo. De una

muestra total de 837 sujetos, se seleccionaron 497 para este estudio, filtrando aquellos casos con
información ante-mortem poco fiable o condiciones de conservación inadecuadas. Dado que la edad
de la muerte puede estimarse a partir de ambas lateralidades (o lados) de la sínfisis púbica, el número
final de muestras es de 986, considerando sujetos en el rango de edad entre 18 y 60 años. Así pues, la
variable de salida de nuestros modelos será directamente el valor numérico de la edad de la muerte
asociado a cada sínfisis púbica, que se conoce en la muestra considerada.

Tabla 1: Rasgos característicos de la sínfisis púbica y valores categóricos asignados

| Variable Name | Categorical Values (Numerical Values) | |
|---|---|---|
| $x_1$ **Articular Face** | Regular Porosity (1) | Ridges Formation (2) |
| | Ridges and Grooves (3) | Grooves Shallow (4) |
| | Grooves Remains (5) | No Grooves (6) |
| $x_2$ **Irregular Porosity** | Absence (1) | Medium (2) |
| | Much (3) | |
| $x_3$ **Upper Symphysial Ext.** | Not Specified (1) | Defined (2) |
| $x_4$ **Bony Nodule** | Absent (1) | Present (2) |
| $x_5$ **Lower Symphysial Ext.** | Not Specified (1) | Defined (2) |
| $x_6$ **Dorsal Margin** | Absent (1) | Present (2) |
| $x_7$ **Dorsal Plateau** | Absent (1) | Present (2) |
| $x_8$ **Ventral Bevel** | Absent (1) | In Process (2) |
| | Present (3) | |
| $x_9$ **Ventral Margin** | Absent (1) | Partially Formed (2) |
| | Formed Without Outgrowths (3) | Formed with Few |
| | Formed With Recesses And Protrusions (5) | Outgrowths (4) |

Los aspectos morfológicos de la sínfisis púbica se analizaron a fondo en la contribución seminal
de Todd [2]. Sin embargo, las descripciones originales eran demasiado genéricas y el proceso de
inspección visual dependía en gran medida de la pericia del antropólogo forense. En un estudio
reciente, Gámez et al. [5] sistematizaron estas descripciones en un intento de reducir el sesgo
subjetivo del análisis. El atlas propuesto definía nueve rasgos asociados al desarrollo y a los cambios
degenerativos de los huesos púbicos. Seguimos este estudio asociando los nueve rasgos (variables $x_1$
a $x_9$) con dos o más valores categóricos, como se muestra en la Tabla 1. Luego transformamos los
valores categóricos en valores numéricos, como se hace en el método de Gilbert-McKern [3], para
diseñar métodos semiautomáticos de la edad de la muerte basados en RS.

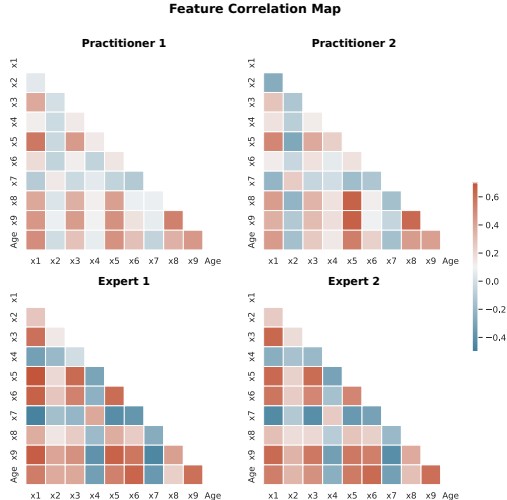

Figura 1: Análisis de correlación (coeficientes de Pearson) para los rasgos de la sínfisis púbica
anotados por cada uno de los observadores.

La muestra fue anotada por cuatro especialistas en AF que etiquetaron a ciegas los nueve rasgos
analizando visualmente cada sínfisis púbica en orden aleatorio. Dos de ellos eran expertos en la
evaluación de los procesos de desarrollo de la sínfisis púbica, mientras que los otros dos eran jóvenes
profesionales con un máster y formación en el proceso de análisis. En un intento de identificar las
relaciones entre los datos etiquetados por cada especialista, en la Fig. 1 se analizan los coeficientes de

159 correlación de Pearson entre la edad real de la muerte de los sujetos y los rasgos anotados. Este análisis
160 muestra claramente la incertidumbre y el sesgo subjetivo incurrido durante el proceso de evaluación.
161 La Fig. 1 también pone de manifiesto el diferente nivel de competencia de los observadores.

162 Para evaluar el rendimiento de los métodos, los datos se particionaron en conjuntos de entrenamiento
163 y de prueba siguiendo una división 80/20, seleccionando la misma muestra para cada anotación de
164 especialista (experto (E) o profesional (P)). La Fig. 2 ilustra la cantidad de muestras consideradas
165 para entrenar los modelos de cada observador y la agregación de datos en bruto (enfoque de fusión de
166 datos) para evaluar la robustez del enfoque según los niveles de especialización de los anotadores.

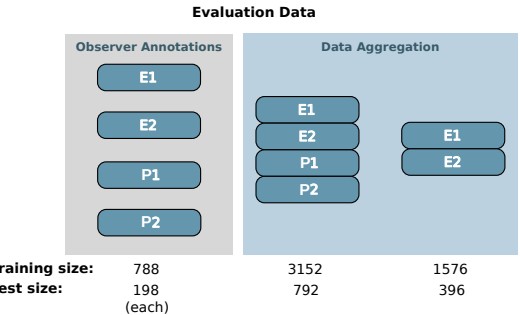

Figura 2: Enfoque de partición de datos. Bloque izquierda: datos utilizados para el aprendizaje de
modelos base de los cuatro anotadores. Bloque derecha: agregación de las anotaciones *Todos* y (sólo)
*Expertos* en un enfoque de fusión de datos mediante aprendizaje conjunto.

167 **3.2.  Diseño del método de Aprendizaje Automático basado en Programación Genética**

168 En esta sección describimos las características específicas del diseño algorítmico de PG como modelos
169 de regresión para identificar relaciones funcionales entre los rasgos óseos anotados [11]. La PG
170 comparte los conceptos principales de los algoritmos genéticos, operando sobre una población de
171 soluciones candidatas que representan expresiones matemáticas potenciales. La población se inicializa
172 con soluciones aleatorias de tamaño pequeño y estas soluciones evolucionan iterativamente utilizando
173 operadores genéticos como el cruce y la mutación.

174 *Esquema de codificación:* Las expresiones matemáticas se codifican en árboles de expresión, cuyos
175 nodos terminales son una de las variables que representan los nueve rasgos identificados ($x_1$ a
176 $x_9$, véase la Tabla 1) o una constante numérica. Los nodos internos del árbol son los operadores
177 matemáticos suma, resta, multiplicación, división, negación e inversa. Se seleccionó un conjunto
178 de operadores sencillos para garantizar que los modelos obtenidos fueran interpretables por los
179 antropólogos forenses. El resultado de la expresión matemática es el valor predicho de la edad de la
180 muerte para los rasgos de la sínfisis púbica especificados como entradas.

181 *Mecanismo de selección:* La PG sigue el esquema generacional clásico tanto para los procedimientos
182 de selección como de reemplazo. Se crea una población intermedia mediante selección por torneo y
183 la población descendiente sustituye directamente a la población de padres considerando el elitismo.
184 Se eligió un torneo de gran tamaño para mantener una alta presión selectiva y, al mismo tiempo,
185 preservar la diversidad en la población. Como se sabe que la PG muestra una convergencia lenta,
186 consideramos tanto un gran tamaño de población como un gran tamaño de torneo para aumentar la
187 capacidad de explotación (véase la Tabla 2).

188 *Operadores genéticos:* Se considera el cruce GP habitual [18], donde se selecciona una arista
189 aleatoria en cada padre, intercambiando ambos subárboles para formar un descendiente en la siguiente
190 generación. Para mantener la diversidad se utilizan dos operadores de mutación diferentes: a) mutación
191 de subárbol, selección aleatoria de una arista y generación aleatoria de un nuevo subárbol que sustituye
192 al antiguo situado en esa arista, y b) mutación puntual, cambio aleatorio del valor de un nodo por otro
193 del mismo tipo: bien un valor numérico aleatorio o bien otra variable/operador.

194 *Función de fitness:* Se basa en el error cuadrático medio (ECM) de la expresión codificada:

$$ECM = \sum_{i=1}^{N}(y_i - y_i')^2 \quad,$$

Tabla 2: Hiperparámetros de la PG después del ajuste

| Parameter | Value |
|---|---|
| Número de generaciones | 1000 |
| Tamaño de población | 1000 |
| Tamaño del torneo | 80 |
| Probabilidad de cruce | 0.75 |
| Probabilidad de mutación de subárbol | 0.11 |
| Probabilidad de mutación puntual | 0.05 |
| Probabilidad de mutación de Hoist | 0.08 |
| Coeficiente de parsimonia | 0.05 |

donde $y_i$ es la edad de la muerte del ejemplo $i$ e $y_i'$ es el valor predicho por el modelo.

*Bloating:* El *bloat* es una limitación bien conocida de la PG, que provoca a un crecimiento excesivo en términos de tamaño de los individuos. El *bloat* conduce a un sobreajuste de los datos y a mayores costes de evaluación debido a la profundidad variable de los árboles. La presión de parsimonia [18,19] es probablemente el método de control más frecuente, que penaliza explícitamente los programas más grandes disminuyendo su fitness proporcionalmente a su tamaño. También consideramos dos alternativas adicionales: i) el enfoque dinámico para calcular el coeficiente de parsimonia de acuerdo con la varianza de los tamaños de los programas dentro de la población en una generación introducida por Poli et al. [20]; y ii) un tercer tipo de mutación, la mutación *hoist* [21], que elimina material genético de la solución padre seleccionando y eliminando aleatoriamente un subárbol.

*Ajuste de parámetros:* Los hiperparámetros se ajustaron sobre un subconjunto de los datos donde otro experto proporcionó un conjunto adicional de anotaciones. En concreto, se consideró el estimador Parzen estructurado en árbol [22] como algoritmo de búsqueda por su buen rendimiento en la optimización de hiperparámetros frente a otros enfoques como la búsqueda aleatoria o en grid. Se basa en la optimización bayesiana, utilizando un modelo probabilístico para encontrar los parámetros óptimos de acuerdo con una función objetivo. Se siguió una validación cruzada de 5 subconjuntos durante la fase de ajuste y la puntuación de validación cruzada se utilizó para guiar la optimización. La Tabla 2 recoge la configuración final de parámetros obtenida.

### 3.3. Diseño de los modelos de conjunto

Dada la complejidad y la incertidumbre inherentes al problema, consideramos el concepto de comité de especialistas con experiencia variada como medio para alcanzar una solución mejorada con respecto a la obtenida por un único profesional. Este concepto está bien establecido en Inteligencia Artificial en forma de aprendizaje conjunto [12], que permite la combinación estratégica de múltiples modelos para resolver un problema de ML mejorando su rendimiento individual.

Concretamente, seguiremos el enfoque habitual de tres etapas [13] para la fusión de información a través de comités basados en modelos independientes, también conocidos como modelos base (véase la Fig. 3). En primer lugar, consideramos una etapa de generación basada en datos. Se entrenarán cuatro modelos base mediante PG en todo el espacio de características utilizando cada conjunto de anotaciones por separado. Para establecer una referencia inferior, compararemos su rendimiento con los modelos entrenados utilizando datos agregados (es decir, todo el conjunto de datos compuesto por cada anotación de los cuatro especialistas o sólo de los dos expertos), como se muestra en la Fig. 2.

La segunda etapa corresponde a la selección o poda del modelo [23]. Para evaluar el impacto del nivel de pericia en la anotación visual de los rasgos óseos, seguiremos dos enfoques: sin poda y poda por pericia. En este último caso, en la tercera etapa sólo se tienen en cuenta los modelos generados a partir de las observaciones de los anotadores expertos.

La última etapa es la de consenso o toma de decisiones, que se consigue combinando las salidas de los modelos base. La fusión de modelos suele hacerse con operadores de agregación como el máximo, la media aritmética o el voto mayoritario [23]. Aunque estos operadores son más eficaces cuando los errores de los modelos base son independientes, esta condición no suele estar garantizada [24].

Proponemos cuatro enfoques distintos para la etapa toma de decisiones del modelo de conjunto considerando diferentes funciones de agregación [14]:

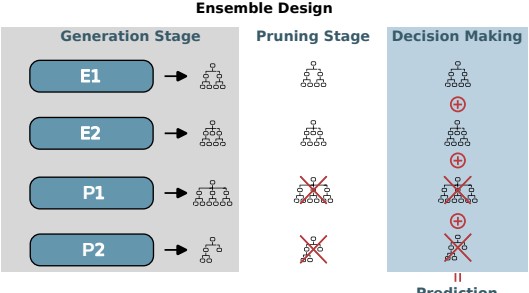

Figura 3: Representación de la metodología de diseño de modelos de conjunto en tres etapas.

- Media simple (MS): Como aproximación naive, los modelos base de RS ($M_i$) se combinan utilizando el operador de media aritmética:

$$MS = \frac{1}{n} \sum_{i=1}^{n} M_i$$

- Media ponderada (MP): Para considerar el nivel de experiencia de los observadores, se asignan pesos manualmente para potenciar la contribución de los modelos base entrenados sobre anotaciones realizadas por expertos. Así, a los *modelos de expertos* se les asigna el doble de peso que a los *modelos de jóvenes profesionales*.

$$MP = \sum_{i=1}^{n} w_i M_i \text{ , where } w_{E_i} = 2 \cdot w_{P_i} \text{ , } \sum_{i=1}^{n} w_i = 1$$

- Media ponderada ordenada (MPO): Una ligera variación consiste en ordenar cada modelo en función de su rendimiento. En lugar de asignar manualmente las ponderaciones, en este enfoque las ponderaciones se asignan proporcionalmente según el coeficiente de determinación de entrenamiento del modelo. $R^2$ relaciona la suma de residuos al cuadrado ($SS_{res}$), calculada como las diferencias entre los valores observados y predichos, con la suma total de cuadrados ($SS_{tot}$), calculada como las diferencias entre los valores observados y la media de los valores observados:

$$R^2 = 1 - \frac{SS_{\text{res}}}{SS_{\text{tot}}}$$

- Agregación fuzzy (AGF): Las agregaciones fuzzy son adecuadas para extraer conocimiento y hacer frente a la incertidumbre del modelo. Algunas de las más populares son la integral de Choquet, la integral de Sugeno y la agregación ponderada ordenada (OWA) [14], considerada en este estudio. La OWA se diferencia de los enfoques anteriores por realizar una agregación ponderada en la que los modelos base $M_i$ se ordenan descendentemente por rendimiento y los pesos se asocian según ese orden de la siguiente forma:

$$w_i = Q_{a,b}\left(\frac{i}{n}\right) - Q_{a,b}\left(\frac{i-1}{n}\right) \text{ , donde}$$

$$Q_{a,b}(x) = \begin{cases} 0, & if \, x < a \\ 1, & if \, x > b \\ \frac{x-a}{b-a}, & otherwise \end{cases}$$

Los vectores de peso para la función OWA se obtuvieron empleando los parámetros $a = 0.3$ y $b = 0.8$.

## 4. Experimentos y Análisis de Resultados

En esta sección realizamos un análisis de rendimiento de los cuatro modelos base, cada uno entrenado con las anotaciones específicas de cada especialista. Comparamos sus resultados con modelos entrenados con anotaciones combinadas siguiendo un enfoque de fusión de información. La Tabla 3 recoge

Tabla 3: Resultados de modelos de referencia y aprendizaje conjunto al estimar la edad de la muerte.

| Modelo | | | EAM | ERCM |
|---|---|---|---|---|
| **Modelos Base** | | E1 | 6.61 | 8.15 |
| | | *E2* | *6.19* | *7.76* |
| | | P1 | 7.69 | 9.20 |
| | | P2 | 7.39 | 8.91 |
| **Agregación de Datos** | | Todos | 7.05 | 8.60 |
| | | *Expertos* | *6.39* | *7.95* |
| **Modelos de Conjunto** | Sin Poda | MS | 6.56 | 7.82 |
| | | MP | 6.35 | 7.65 |
| | | *MPO* | *6.27* | *7.59* |
| | | AGF | 6.43 | 7.76 |
| | Con Poda | MS | 6.14 | 7.54 |
| | | MP | 6.11 | 7.52 |
| | | **MPO** | **6.08** | **7.52** |
| | | AGF | 6.11 | 7.53 |

los resultados obtenidos por cada modelo en la experimentación para dos medidas de evaluación: el error absoluto medio (EAM) y el error de raíz cuadrada media (ERCM), diferentes de la función de fitness utilizada para el proceso de aprendizaje, el ECM (véase la Sec. 3.2).

Los resultados de los modelos base subrayan la importancia de los niveles de competencia de los antropólogos forenses. Aunque el sesgo subjetivo sigue presente en el proceso de anotación de rasgos, los modelos entrenados con etiquetas de expertos superaron a los basados en etiquetas de jóvenes profesionales en aproximadamente 1 año tanto en EAM como en ERCM. El mejor resultado es el del modelo diseñado con las anotaciones del segundo experto (E2, resaltado en cursiva en la tabla).

Esas diferencias se propagan también a los modelos que consideran un enfoque de fusión de información, mostrando un rendimiento cercano a la media de los cuatro modelos base. La incorporación de las anotaciones de los expertos ayuda a mejorar el rendimiento de los modelos base de los profesionales, mientras que la consideración conjunta de las anotaciones de ambos expertos no mejora los resultados individuales del modelo base E2.

Analizando los modelos de conjunto que consideran todos los modelos base generados (es decir, sin estrategia de poda). En general, las cuatro variantes presentan un rendimiento competitivo, destacando la MPO como la agregación con el mejor rendimiento (en cursiva). Sin embargo, es el único modelo de conjunto no podado capaz de superar al mejor modelo base (E2) y sólo en la medida ERCM.

Concluimos que la incertidumbre y las anotaciones potencialmente erróneas de los observadores profesionales (con nivel de máster) penalizan el rendimiento de los modelos generados. Confirmamos esta suposición mediante los modelos de conjunto podados que solo consideran las anotaciones de los dos expertos, los cuales fueron capaces de proporcionar resultados robustos, mejorando los modelos base, independientemente del operador de agregación considerado. De nuevo, la MPO destaca como el mejor enfoque al obtener los errores de predicción más bajos, EAM=6.08 años y ERCM=7.52 años, que constituyen un estado del arte ya que [5] obtuvo MAE=13.19 y ERCM=10.38 en error de prueba para un conjunto de pubis con el mismo rango de edad (18-60 años) y [4] obtuvo EAM=12.1 y ERCM=9.7 en error de entrenamiento para un rango de edad de 19-100 años.

La Fig. 4 muestra la distribución de las estimaciones de los cuatro modelos base y los dos modelos de conjunto con agregación MPO (podados y no podados). Identificamos un patrón claro: se sobreestima la edad en los sujetos jóvenes y se subestima en los sujetos mayores. Este comportamiento es un problema bien conocido en el área [25]. Los resultados demuestran la relevancia del nivel de competencia y la fiabilidad de las anotaciones de los especialistas forenses. Las diferencias entre observadores son más acusadas en el caso de los más novatos, dado que los modelos entrenados con las etiquetas de los expertos tienden a proporcionar resultados más precisos y una menor variabilidad, aunque no existe un consenso aparente entre ambos expertos. Las etiquetas inexactas penalizan fuertemente las predicciones conjuntas, especialmente en el menor rango de edad (18-23 años). No obstante, la fusión de las predicciones de los expertos ha sido un enfoque acertado, que permitió al modelo de conjunto ofrecer estimaciones más precisas que los modelos individuales.

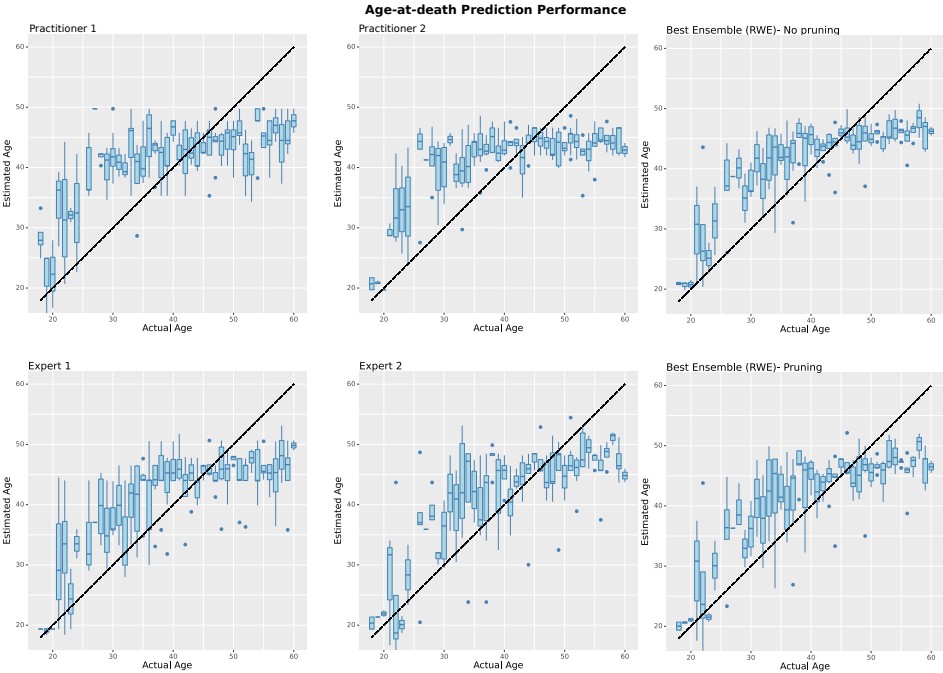

Figura 4: Análisis visual del rendimiento de varios modelos. Cada boxplot muestra el error de estimación para cada edad. La línea de puntos negra es la referencia de la estimación real.

## 5. Conclusiones

En este artículo hemos explorado un enfoque de fusión de información de múltiples anotadores de características en una tarea compleja de ML caracterizada por incertidumbre y sesgo subjetivo. Para abordar el problema de la estimación de la edad de la muerte a partir de la sínfisis púbica adoptamos un enfoque de RS utilizando PG. Este EA produce expresiones matemáticas interpretables y fácilmente aplicables, asistiendo a los antropólogos forenses en el análisis.

Hemos considerado el aprendizaje conjunto y la toma de decisiones empleando diferentes operadores de agregación, logrando una mejora del rendimiento y aumentando la robustez de las predicciones en comparación con los modelos individuales. Siguiendo este enfoque, conseguimos tratar adecuadamente la incertidumbre inherente al proceso de anotación de muestras por parte de los antropólogos forenses, necesario para el diseño del método semiautomático de estimación de la edad de la muerte.

Obtuvimos dos conclusiones relevantes. El sesgo subjetivo de la evaluación visual para determinar los rasgos óseos está estrechamente relacionado con la experiencia de los especialistas humanos. Así, puede dar lugar a anotaciones poco fiables, que dificultan la agregación de datos de múltiples especialistas. No obstante, el uso de modelos de conjunto nos permitió armonizar diversas observaciones de antropólogos forenses muy competentes, mejorando el rendimiento de los modelos base.

## Agradecimientos

Esta investigación ha sido subvencionada por el proyecto CONFIA (PID2021-122916NB-I00) financiado por MCIN/AEI/10.13039/501100011033 y por "FEDER: Una manera de hacer Europa".

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
