# OpenReview forum: "Estimación de la edad de la muerte basada en aprendizaje conjunto y regresión simbólica a partir de múltiples anotaciones"
_MAEB/2025/Congreso — MAEB 2025_

### Official Review · Reviewer_eA8c · 2025-03-14
**Aplicación de programación genética y regresión simbólica al problema propuesto**

**Rating:** 5
**Confidence:** 5

**Review:**

Interesante problema abordado en el que se aplica regresión simbólica mediante programación genética para estimar la edad a partir de la sínfisis púbica.  Utilizan una base de datos fiable y completa para el aprendizaje.

La metodología está bien planteada, y los resultados de interesantes, aunque se pueden plantear cuestiones para trabajo futuro:  ¿Otros métodos de control de Bloat?  ¿Modelos estructurados para mejorar los resultados de GP?

---

### Official Review · Reviewer_EQSH · 2025-03-14
**Muy buen trabajo, con gran interés práctico**

**Rating:** 5
**Confidence:** 5

**Review:**

El artículo presenta la aplicación de regresión simbólica a través de programación genética para la estimación de la edad del fallecimiento a partir de datos óseos. Los resultados toman como base un conjunto de datos generado por 4 antropólogos, 2 expertos y 2 de nivel máster, que se combinan con diferentes tipos de agregación.

Los resultados de la PG mejoran los de los expertos, aunque la afirmación que indica que representa el estado del arte depende de aclarar si [5] y [4] usaron exactamente los mismos bancos de datos. En todo caso, son unos resultados excelentes, y la investigación es muy relevante.

El artículo está muy bien estructurado y redactado, por lo que solamente se incluyen algunos comentarios beves:

La descripción de la figura 1 es mejorable. Se indica que muestra la correlación entre la edad real de la muerte y "los rasgos anotados". ¿A qué se refiere "los rasgos anotados"? ¿Al número de rasgos presentes? (algunos tienen diferentes categorías).

Dado que el artículo se escribe en español, se recomienda que las tablas y las figuras de elaboración propia se muestren en español. La tabla 2, por ejemplo, quedó con encabezados en inglés.

Erratas:

Pg. 3: "Los métodos … edad de la muerte son obsoletos" -> "están obsoletos"

Pg. 8: El modelo MP con poda obtiene un ERCM igual que el mejor, por lo que debería ir en negrita, a menos que se deba a los decimales que no vemos (por lo que habría que comentarlo).

Pg. 8: MAE -> EAM

---

### Official Review · Reviewer_1hMb · 2025-03-17
**Estimación de la edad de la muerte**

**Rating:** 5
**Confidence:** 4

**Review:**

La estimación de la edad de la muerte es una tarea compleja con importantes aplicaciones en Medicina Forense. Uno de los enfoques más usados, propuesto inicialmente por Todd y mejorado posteriormente por Gilbert y McKern, consiste en estimar la edad de la muerte a partir de las características de la sínfisis púbica anotadas por un panel de forenses. Cada forense anota un valor, en el rango [1, 5], a cada rasgo del hueso. Estos valores son posteriormente agregados para obtener una estimación de la edad de la muerte.
En este trabajo se propone un procedimiento de estimación semiautomática que sigue el esquema anterior. El panel de forenses está formado por cuatro especialistas, dos de ellos con una amplia experiencia y otros dos con una experiencia limitada en el proceso de observación y anotación de los rasgos de la sínfisis púbica. Se emplea la Regresión Simbólica y la Programación Genética para obtener un modelo de predicción.

La experiencia computacional analiza el efecto que tiene el grado de experiencia de los especialistas y la función de agregación empleada en la calidad del modelo de predicción obtenido. Se concluye que las anotaciones realizadas por los especialistas con menor experiencia penalizan el rendimiento del modelo y que la agregación de las anotaciones mejora a los modelos base. El enfoque basado en MPO obtiene estimaciones más precisas que las suministradas por métodos de la literatura.

El trabajo está muy bien escrito y estructurado. Además, la experiencia computacional desarrollada muestra la calidad y validez de la propuesta y de su utilidad para ser integrada en un sistema semiautomático de predicción de la edad de la muerte.

Recomendaciones:

- Añadir alguna referencia bibliográfica de Regresión simbólica.
- Línea 120, cambiar EAs, por AEs.
- Línea 302, cambiar EA por AE.

---

### Decision · Program_Chairs · 2025-03-20

Accept